# The Prediagnostic General Practitioners’ Pathway of Gastrointestinal Stromal Tumor Patients: A Real-World Data Study

**DOI:** 10.3390/cancers17091391

**Published:** 2025-04-22

**Authors:** Emily I. Holthuis, Verena Slijkhuis, Winette T. A. van der Graaf, Cas Drabbe, Winan J. van Houdt, Yvonne M. Schrage, Tim C. Olde Hartman, Annemarie Uijen, Neeltje Steeghs, Isabelle Bos, Marianne Heins, Olga Husson

**Affiliations:** 1Department of Medical Oncology, Netherlands Cancer Institute, 1066 CX Amsterdam, The Netherlandsw.vd.graaf@nki.nl (W.T.A.v.d.G.); n.steeghs@nki.nl (N.S.); o.husson@nki.nl (O.H.); 2Department of Medical Oncology, Erasmus MC Cancer Institute, Erasmus University Medical Center, 3015 GD Rotterdam, The Netherlands; 3Nutrition and Disease Department, Wageningen University and Research (WUR), 6708 PB Wageningen, The Netherlands; 4Radboud University Medical Center, Radboud Institute of Medical Innovation, Department of Primary and Community Care, 6525 GA Nijmegen, The Netherlands; cas.drabbe@radboudumc.nl (C.D.); tim.oldehartman@radboudumc.nl (T.C.O.H.); annemarie.uijen@radboudumc.nl (A.U.); 5Surgical Oncology Department, NKI-AVL—Netherlands Cancer Institute/Antoni van Leeuwenhoek, 1066 CX Amsterdam, The Netherlands; w.v.houdt@nki.nl (W.J.v.H.); y.schrage@nki.nl (Y.M.S.); 6Department of Medical Oncology, University Medical Center Utrecht, 3584 CX Utrecht, The Netherlands; 7Netherlands Institute for Health Services Research (Nivel), 3513 CR Utrecht, The Netherlands; i.bos@nivel.nl (I.B.); m.heins@nivel.nl (M.H.); 8Department of Surgical Oncology, Erasmus MC Cancer Institute, Erasmus University Medical Center, 3015 GD Rotterdam, The Netherlands; 9Department of Public Health, Erasmus University Medical Center, 3015 GD Rotterdam, The Netherlands

**Keywords:** gastrointestinal stromal tumor, soft-tissue sarcoma, real-world data, primary healthcare, case-control

## Abstract

Gastrointestinal stromal tumors (GISTs) are rare digestive tract cancers that are often challenging to diagnose early due to nonspecific symptoms. General practitioners (GPs) are typically the first to evaluate these symptoms, making them crucial for early detection. However, limited research has examined how GIST patients seek medical care before diagnosis. This study investigates GP visits, reported diagnoses, and prescribed medications in the year preceding a GIST diagnosis. Our findings indicate that GIST patients attend more frequent GP visits, with a significant increase in the months leading up to diagnosis, often due to digestive issues and anemia. Future research should analyze GP records in greater detail to refine strategies for earlier recognition and intervention, eventually leading to faster diagnosis and better patient outcomes.

## 1. Introduction

Gastrointestinal stromal tumors (GISTs) are the most prevalent mesenchymal tumors within the gastrointestinal (GI) tract, but account for only 2% of all GI tumors [1,2]. Despite this, GISTs are considered rare, with an incidence of 1.5 per 100,000 people annually [3]. GISTs are primarily caused by activating mutations in KIT or PDGFRα oncogenes [4,5,6] and most often develop in the stomach (56%) and small intestine (32%), with rarer occurrences in the colorectum, esophagus, and appendix [1,2,7]. Surgical resection is the cornerstone treatment for resectable GISTs. Surgery alone can be curative for approximately half of these patients, with 5-year relapse-free survival rates ranging from 70 to 80%, depending on whether adjuvant Imatinib is administered [8].

GISTs present heterogeneously, with 18% of cases being asymptomatic [7,9]. Symptoms depend on tumor size, location, and growth pattern, and often include GI bleeding with subsequent anemia and abdominal pain. Other symptoms may include dyspepsia, weakness, distension, weight loss, melena, hematemesis, GI obstruction, and early satiety [9,10,11,12]. Small GISTs (<2 cm) are frequently discovered incidentally [12,13]. Currently, in cases without metastases, complete surgical resection of GISTs is the only possible curative treatment, underlining the importance of early diagnosis [14,15,16]. However, vague symptoms often delay initial diagnosis, with approximately 20% of patients presenting with metastases to the peritoneum or liver at the time of diagnosis [17]. Limited awareness among health care professionals further contributes to diagnostic delays [18,19].

In the Netherlands, almost all inhabitants are registered with a general practitioner (GP), who is the point of referral to secondary care [20]. GPs manage >95% of presented medical problems and refer patients to secondary care when needed [20]. Hence, they are often the first point of contact for potential early signs of GISTs. Since GPs see a large percentage of patients before diagnosis, it could be useful to examine the prediagnostic trajectory and identify diagnoses made by GPs prior to a cancer diagnosis. This can be achieved with data registered by GPs in electronic health records (EHRs). Hence, real-world data (RWD), which refers to data collected outside a clinical trial setting, from sources like EHRs, have been shown to be relevant for research on diagnostic delays in sarcoma patients, and are being increasingly utilized [21]. For instance, previous research has shown that sarcoma patients experience multiple prediagnostic contacts in primary care, resulting in delays in referral to specialized care and time to diagnosis [22,23,24]. However, few studies focus specifically on GISTs, often with limited sample sizes [19]. Given the diverse clinical presentation of GISTs, examining and understanding the reasons for and frequency of GP consultations before a GIST diagnosis could offer insights into the primary care pathway of this group [25,26,27,28].

The objective of this study, therefore, is to investigate the number of GP contacts, the diagnoses registered during these contacts, and the type of prescribed drugs in primary care during the 12 months preceding GIST diagnosis.

## 2. Materials and Methods

### 2.1. Netherlands Cancer Registry

The Netherlands Cancer Registry (NCR), hosted by the Netherlands Comprehensive Cancer Organization (IKNL), has recorded diagnostics, diagnosis, tumor characteristics, and initial treatment of all cancer patients treated in Dutch hospitals since 1989 [29].

### 2.2. Nivel-PCD

The Netherlands Institute for Health Services Research (Nivel) collects longitudinal EHR data from GPs. Since 2012, these data have been processed into the Nivel Primary Care Database (Nivel-PCD), which contains data on contacts, diagnoses, and referrals to secondary care from around 500 GPs across the Netherlands, covering approximately 10% of the Dutch population [30,31]. Almost all Dutch inhabitants are registered with a GP [32], and Nivel-PCD patients form a representative sample of the Dutch population, based on age and sex [30]. The study was approved under the Nivel-PCD governance code NZR-00321.069.

### 2.3. Linkage Procedure and Study Population

Data from the NCR and Nivel-PCD were linked by a trusted third party (ZorgTTP) using pseudonyms based on date of birth, sex, and four-digit postal code [33].

Adults (≥18 years) diagnosed with GISTs (ICD-10-GM codes C49.A0-C49.A5 and C49.A9) between 1 January 2010 and 31 December 2020 were identified within the NCR, linked to the Nivel-PCD data, and included in the study. Patients with other types of soft-tissue or bone sarcomas were excluded and presented in a separate paper [24].

For comparison, cases were matched to controls from Nivel-PCD in a 1:2 (n = 282) or 1:1 (n = 12) ratio, depending on available controls. Controls, without a cancer diagnosis, were matched with respect to age (±5 years), sex, GP practice, and registration years. Each matched control was assigned the same diagnosis date as their corresponding GIST case, referred to as the index date.

For clarity, the months before diagnosis are referred to as ‘Time x’, where ‘Time 2’ represents two months prior to the diagnosis.

### 2.4. Characteristics

Sociodemographic characteristics (age, sex) and tumor characteristics (e.g., diagnosis date, primary tumor site, grading) were obtained from the NCR.

### 2.5. General Practice Care

Primary healthcare data were extracted from the Nivel-PCD database. GPs register diagnoses and symptoms during contacts in EHRs, using the International Classification of Primary Care (ICPC) and medication prescriptions coded according to the World Health Organization (WHO) Anatomical Therapeutic Chemical (ATC) classification system [30]. The database allows a maximum of three ICPC codes per contact.

The number of contacts with the GP, GP assistant, or general practice mental health professional (face-to-face contacts and phone contacts) was extracted for the 12-month period prior to diagnosis or the index date. Furthermore, all information on diagnosis and symptoms registered by the GP at the time of the contact, as well as all prescribed drugs, was included for the 12 months prior to diagnosis or the index date. Data on referral to secondary care and on relevant diagnostic procedures, such as CT and endoscopy, were not available.

### 2.6. Statistical Analysis

Descriptive statistics were used to summarize baseline and tumor characteristics. Categorical data are presented in numbers (n) and percentages (%), and continuous data as means (SD) or medians (IQR), depending on the distribution. The mean number of monthly GP contacts was calculated by dividing the number of GP contacts in each month by the total number of GIST cases or cancer-free controls for the 12 months prior to diagnosis or the index date. The Wilcoxon rank-sum test was used to compare monthly contacts between cases and controls. This non-parametric test was chosen to examine differences between the two independent groups and no adjustments were made for potential confounders, as this was not the primary aim. Common health conditions were identified by ICPC codes associated with each contact, and the most common health conditions are presented as numbers (n) and percentages (%). Newly prescribed drugs in the 12 months prior to diagnosis/index date were assessed, identifying only new prescriptions based on a 12-month run-in period and the fourth ATC code level [34]. Commonly prescribed drugs are reported as counts and percentages. The most prevalent health conditions and newly prescribed drugs were analyzed in two intervals: 0–4 and 5–12 months. Sensitivity analyses evaluated GP contact frequency over the 24 months prediagnosis. Statistical significance was set at *p* ≤ 0.05, with analyses conducted in R (RStudio version 4.0.2).

### 2.7. Privacy

Electronic health records can be used for research without informed consent or ethics committee approval when no directly identifiable data are involved (Art. 24 GDPR Implementation Act jo Art. 9.2 sub j GDPR).

## 3. Results

A total of 298 patients were diagnosed with a GIST between 1 January 2010 and 31 December 2020 (Figure 1). Four GIST cases were excluded due to the lack of available controls. Subsequently, 282 pairs of cases and controls were matched in a 1:2 ratio, and 12 pairs of cases and controls were matched in a 1:1 ratio, resulting in 576 matched controls. GIST cases (N = 294) and controls (N = 576) were similar in age (66.44 years ± SD 12.76 vs. 66.85 ± SD 12.94) and gender distribution (47.96% females vs. 47.74% males) (Table 1). Among the GIST cases, 66.7% were diagnosed with gastric GISTs, 33.0% with non-gastric GISTs, and one case (0.3%) presented with an unspecified GIST location.

The median number of GP visits among the investigated cases in the 12 months preceding diagnosis was six (IQR 4–11), compared to three (IQR 2–6) for cancer-free controls. This difference was statistically significant (*p* < 0.05). Statistically significant differences in the mean number of monthly GP contacts between cases and controls were observed in the months closer to diagnosis, from Time 4 to Time 1 (Figure 2). However, cases showed a noticeably higher number of contacts starting from Time 7. In contrast, the mean number of monthly GP contacts for controls remained stable throughout the 12-month period. A sensitivity analysis over 24 months showed stable average monthly GP contacts for both groups in the Time 24–Time 12 period (Figure A1).

Table 2 presents the most common health conditions registered by GPs for GIST patients (cases), along with their prevalence among cancer-free controls. In the 4 months before diagnosis, the most common health conditions in GIST cases were malignant neoplasm of the stomach (D74) (27.9%) and malignant neoplasm of other or unspecified digestive sites (D77) (27.6%). Other GI conditions included malignant neoplasm of the colon/rectum (D75) (11.2%), other localized abdominal pain (D06) (9.5%), generalized abdominal pain/cramps (D01) (7.5%), and stomachache/stomach pain (D02) (7.1%). Iron deficiency anemia (B80) was also common (9.5%) (Table 2).

All health conditions, except for malignant neoplasm of the stomach (D74) and malignant neoplasm of other digestive organs/NOS (D77), were also present in the controls. Cystitis/other urinary infection (U71), uncomplicated hypertension (K86), and diabetes mellitus (T90) were more common in controls, while malignant neoplasm of the colon/rectum (D75) (11.2% vs. 2.6%), iron deficiency anemia (B80) (9.5% vs. 0.2%), other localized abdominal pain (D06) (9.5% vs. 0.5%), generalized abdominal pain/cramps (D01) (7.5% vs. 0.2%), and stomachache/stomach pain (D02) (7.1% vs. 0.5%) were more frequently observed in the cases. In the 5–12 months leading up to diagnosis, prevalence of most health conditions were similar between cases and controls, except for other localized abdominal pain (D06) (6.8% vs. 0.5%), chronic ulcer skin/bedsore (S97) (6.1% vs. 0.2%), malignant neoplasm stomach (D74) (5.8% vs. 0%), stomach ache/stomach pain (D02) (5.4% vs. 0.5%) and leg/thigh symptoms/complaints (L14) (5.4% vs. 0.7%) (Table 2).

In the 4 months prior to diagnosis, the most frequently prescribed drugs for GIST cases were osmotically acting laxatives (A06AD) (15.0%) and proton pump inhibitors (A02BC) (13.9%). Other commonly prescribed drugs included propulsives (prokinetics) (A03FA) (5.8%), contact laxatives (A06AB) (3.7%), iron bivalent oral preparations (B03AA) (3.4%), and benzodiazepine derivatives (N05BA) (3.1%). These drugs, except for iron bivalent oral preparations (B03AA), were also prescribed to controls, albeit less frequently (Table 3).

## 4. Discussion

In this case–control study, we aimed to investigate the primary care pathway of GIST patients in the 12 months leading up to diagnosis. An increase in the mean number of monthly GP contacts was observed starting 4 months before GIST diagnosis, peaking 1 month prior, while controls had stable contact rates. This is the first study to specifically investigate the prediagnostic trajectory of GIST patients. A previous Dutch case–control study and a retrospective Australian cohort study reported similar findings, showing that GP contacts were most frequent just before a sarcoma diagnosis [21,23]. In the 6 months preceding diagnosis, patients had a median of three to four GP contacts [23], similar to our findings of six GP contacts in the 12 months prior to diagnosis. Additionally, a patient-reported study showed 41.3% of bone sarcoma and 31.9% of STS patients had three or more GP contacts before referral, indicating middle-to-high diagnostic difficulty [22,29]. The increase in the consultation numbers observed in our study could be attributed to two factors: (1) patients returning (often as agreed) for follow-up visits due to persistent symptoms until a GIST diagnosis was made at a hospital, or (2) patients with alarm symptoms being promptly referred for further investigations, as per Dutch guidelines [30], resulting in a single GP visit. For serious diagnoses, follow-up contact may have become more frequent, prior to and continuing after diagnosis. These data do not clarify which scenario applies, making it difficult to assess if the GP-level diagnostic process could be improved. If a GP required six consultations before making an appropriate specialist referral, there is room for improvement. However, if the GP referred the patient promptly and the increased contacts were follow-ups after a suspicion of malignancy, no improvements at the GP level could be made.

In this study, diagnoses reflecting malignant neoplasms of the GI tract, abdominal pain, and iron deficiency anemia were prevalent among GIST cases, while they were rarely documented among controls, especially in the 4-month period prior to diagnosis. The high number of recorded diagnoses of malignant neoplasms of the GI tract in the primary care database likely reflects instances where ICPC codes were updated as more clinical information became available. For example, a patient may have initially presented with stomach complaints (ICPC code: stomach complaints). Subsequent blood tests may have revealed iron deficiency anemia (ICPC: iron deficiency anemia), leading to a gastroscopy that raised a suspicion of malignancy (ICPC: stomach cancer). Finally, pathological analysis by an oncologist may have confirmed a GIST diagnosis (ICPC code: stomach malignancy), thereby reclassifying the initial diagnosis as a stomach cancer. Notably, no ICPC code specifically categorizes GISTs. Additionally, the timing of data extraction from the GP information system represents a snapshot of the EHR, meaning that the information may have been updated later.

Previous research found that GI bleeding and abdominal pain were common in gastric GISTs, while acute abdominal symptoms were frequent in small intestine GISTs, particularly jejunal and ileal types [31]. However, it was not possible to compare symptoms across GIST locations.

Common symptoms in both GIST cases and cancer-free controls included hypertension, diabetes mellitus, excessive ear wax, cystitis/other urinary infection, and cough. These diagnoses reflect common reasons for GP visits, as patients with hypertension or diabetes typically have routine check-ups.

While we observed a substantial number of symptoms potentially related to GISTs, we cannot rule out other underlying health conditions with similar symptoms. Previous research has noted that primary tumors like gastric and colorectal adenocarcinomas tend to be discovered alongside GISTs, although these findings are based on small sample sizes [32,33,34]. These cancers share overlapping symptoms with GISTs, such as weight loss, persistent abdominal pain, early satiety, rectal bleeding, abdominal mass, and iron deficiency anemia [35,36,37]. Our study did not include comorbidity data for GIST cases. Comparative symptoms of different tumors in the GI tract, as well as benign conditions, could be examined further, given that the symptoms of GISTs may closely resemble those of other tumors or benign lesions, making them less reliable for accurately identifying GISTs [38]. For example, a previous study involving colorectal cancer (CRC) patients identified similar reasons for GP visits, such as diagnoses of malignant neoplasm of the colon/rectum, iron deficiency anemia, and localized abdominal pain [39]. Consequently, diagnosing GISTs remains challenging for GPs due to their diverse clinical presentation, symptom overlap with other GI tumors and benign lesions, and the rarity of the disease [19].

The newly prescribed drugs among GIST cases were also seen among the controls, such as osmotically acting laxatives and proton pump inhibitors, though to a lesser extent. Iron bivalent oral preparations were exclusively prescribed to cases in the 4 months prior to diagnosis. Propulsives and contact laxatives are used to enhance gastrointestinal motility and promote bowel movements and are often prescribed for constipation. These drugs may also be prescribed for more nonspecific abdominal complaints, where constipation is suspected as a possible cause. Iron supplements are commonly given to treat iron deficiency, a common condition among GIST cases. Notably, in the 4 months prior to diagnosis, specific drug prevalence seemed higher among GIST cases compared to the controls, a trend that was less pronounced during the 5–12-month period before diagnosis. A previous study of CRC patients observed similar prescription patterns, including the use of laxatives and proton pump inhibitors supplements [39].

Our study possesses several strengths. The GIST patient sample is representative of the broader GIST population, with the proportion of gastric GISTs closely matching their actual epidemiological distribution [7]. The male-to-female ratio and median age at diagnosis are consistent with previous research [7,40,41,42,43,44]. Furthermore, EHR data, routinely recorded by GPs, reduce recall bias compared to patient-reported outcomes [45]. The use of RWD in the current study included diverse groups, increasing its generalizability potential [21]. However, there are limitations. We could not further stratify by anatomical subtypes of GISTs due to the relatively small sample size. Sociodemographic data other than age and sex were unavailable, and other socio-economic factors such as lower SES, older age, lower educational level, higher prevalence of certain health conditions, and stressful life events are often associated with increased GP visits [46,47]. We also lacked data on lifestyle or environmental risk factors for GISTs, as well as on referrals to secondary care or diagnostic procedures, with such data possibly providing further insights into diagnostic pathways and potential delays. The retrospective design of this study may have introduced some bias. However, we believe that we selected a representative sample of the broader GIST population. Recall bias, which is more common in questionnaire-based studies, was likely minimal, as the data were recorded either during or shortly after patient consultations. However, because primary care data are not collected for research purposes, they may be incomplete—a common limitation of RWD [21]. Furthermore, we were unable to review uncoded diagnoses and symptoms in unstructured data (i.e., clinical information expressed in free text, not restricted to predefined ICPC codes), a fact that may have led to some underrepresentation of observations in primary care. Additionally, the sole usage of primary care data, including consultation frequency, diagnosis, and prescribed drugs, is not enough to map out the prediagnostic trajectory. We did not observe any painkillers in the list of drugs, a fact that can likely be explained by the fact that over-the-counter medications are not registered, and, therefore, we have no information on their use. Additionally, our findings may not be generalizable to healthcare systems where access to care may be more limited. Additionally, there is always some variation among doctors, i.e., inter-doctor variation and potential coding inconsistencies; however, this impact is minimized when comparing cases and controls, as they were matched on, among other things, the same GP practice.

### Future Studies

Future research could explore implementing a system that sends questionnaires to both patients and GPs at the time of diagnosis at the hospital. This would collect information on symptoms experienced prior to diagnosis, what was reported to the GP, and any symptoms not mentioned or reasons for not visiting the GP. Such data could help assess whether GPs recognized and documented these symptoms as relevant. This study highlights the increase in consultations in primary care, highlighting the significant role of the GP in the diagnostic pathway. Investigating the reasons for multiple contacts prior to diagnosis would be valuable in determining if improvements in care can be made at the GP level or if the focus should shift toward secondary and tertiary care. Additionally, a detailed analysis of EHRs, including unstructured data, should be conducted to capture nuanced insights. Natural language processing could help extract meaningful information from unstructured data, while machine learning techniques could identify potential patterns and relationships within the data [48]. Furthermore, future studies should acquire information on laboratory and diagnostic test results, as well as on referral dates to secondary care, to provide a clearer picture of the prediagnostic trajectory [21,49].

## 5. Conclusions

Our findings reveal differences in the mean number of monthly GP visits between GIST cases and cancer-free controls, indicating increased consultations prior to diagnosis. However, the exact reasons behind these increased visits remain unclear. Diagnoses related to malignant GI tract neoplasms, various types of abdominal pain, and iron deficiency anemia were common among GIST cases in the 4 months before diagnosis. However, some diagnostic codes (e.g., malignant GI tract neoplasms) may be updated as more clinical information becomes available, highlighting the need to investigate the initial symptoms. While this study does not assess whether the diagnostic process at the GP level could be improved, it provides a valuable first step toward understanding the prediagnostic journey of GIST patients in primary care. Future research should explore this further, leveraging both EHR data (including free-text entries) and insights directly from patients and GPs.

## Figures and Tables

**Figure 1 cancers-17-01391-f001:**
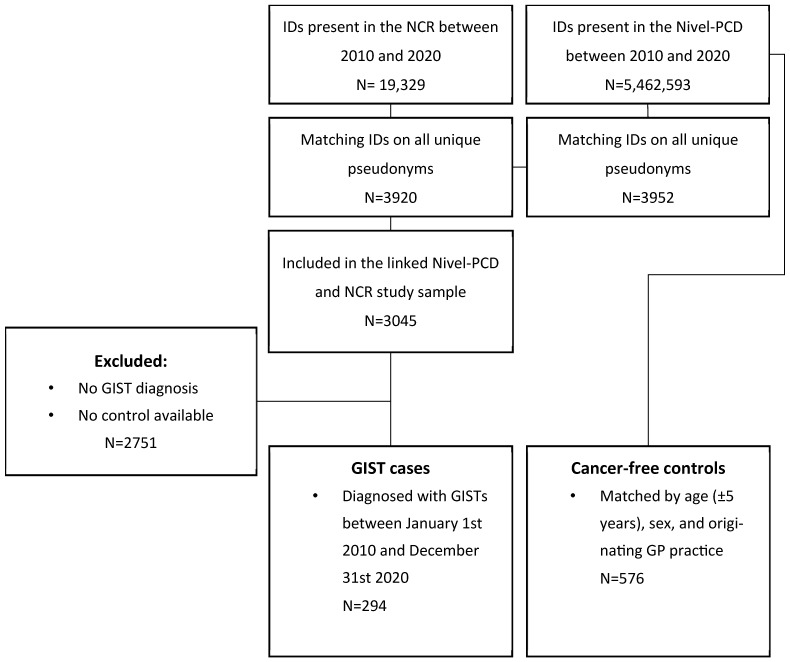
Flowchart of the study population.

**Figure 2 cancers-17-01391-f002:**
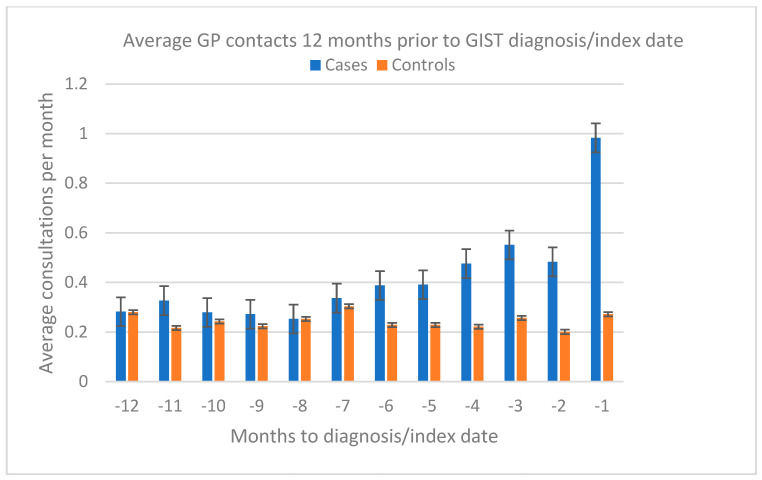
Mean number of monthly GP contacts in the 12 months prior to GIST diagnosis/index date.

**Table 1 cancers-17-01391-t001:** Baseline characteristics of the NCR-PCD sample.

	GIST Cases N = 294	Cancer-Free Controls N = 576
Gender, male, n (%)	141 (48.0)	275 (47.7)
Age at time of diagnosis * (years), mean (SD)	66.4 (12.8)	66.9 (12.9)
GP visits **, median (IQR)	6 (4–11)	3 (2–6)
Year of diagnosis		
2010–2013, n (%)	84 (28.6)	NA
2014–2017, n (%)	116 (39.5)	NA
2018–2020, n (%)	94 (32.0)	NA
GIST subtype		
Gastric GIST, n (%)	196 (66.7)	NA
Gastrointestinal GIST, other (excl. stomach), n (%)	97 (33.0)	NA
Gastrointestinal GIST, other		
Gastric GIST, n (%)	1 (0.34)	NA

* For GIST-free controls: the diagnosis year of the matched GIST case was used to calculate age at diagnosis. ** Number of GP visits in the 12 months preceding diagnosis or the index date.

**Table 2 cancers-17-01391-t002:** The most common diagnoses coded by the GP in the 0–4 and 5–12 months prior to GIST diagnosis (cases) and the index date (controls).

	GIST Cases N = 294		Cancer-Free Controls N = 576
ICPC coded ≤ 4 months prior to diagnosis			
ICPC meaning (ICPC)	n (%)	Days between visit and diagnosis ± SD	n (%)
Malignant neoplasm of the stomach (D74)	82 (27.9)	36.9 ± 33.3	0 (0)
Malignant neoplasm of other digestive organs/NOS (D77)	81 (27.6)	49 ± 40.5	0 (0)
Malignant neoplasm of the colon/rectum (D75)	33 (11.2)	32.3 ± 23.6	15 (2.6)
Iron deficiency anemia (B80)	28 (9.5)	43.4 ± 25.6	1 (0.2)
Other localized abdominal pain (D06)	28 (9.5)	31.6 ± 35.2	3 (0.5)
Cystitis/other urinary infection (U71)	28 (9.5)	58.6 ± 35.7	31 (5.4)
Uncomplicated hypertension (K86)	26 (8.8)	58.1 ± 32.1	35 (6.1)
Diabetes mellitus (T90)	26 (8.8)	60.4 ± 36.2	42 (7.3)
Generalized abdominal pain/cramps (D01)	22 (7.5)	37.0 ± 33.6	1 (0.2)
Stomachache/stomach pain (D02)	21 (7.1)	47.0 ± 34.1	3 (0.5)
ICPC coded ≤ 12–5 months prior to diagnosis			
ICPC meaning	n (%)	Days between visit and diagnosis ± SD	n (%)
Uncomplicated hypertension (K86)	68 (23.1)	237.0 ± 65.5	102 (17.7)
Diabetes mellitus (T90)	36 (12.2)	228.8 ± 65.2	57 (9.9)
Cystitis/other urinary infection (U71)	28 (9.5)	264.0 ± 59.5	33 (5.7)
Other localized abdominal pain (D06)	20 (6.8)	248.4 ± 59.1	3 (0.5)
Excessive ear wax (H81)	19 (6.5)	227.0 ± 79.1	36 (6.3)
Chronic ulcer skin/bedsore (S97)	18 (6.1)	261.0 ± 80.7	1 (0.2)
Malignant neoplasm of the stomach (D74)	17 (5.8)	190.8 ± 44.6	0 (0)
Cough (R05)	17 (5.8)	222.2 ± 68.4	26 (4.5)
Stomachache/stomach pain (D02)	16 (5.4)	166.6 ± 69.5	3 (0.5)
Leg/thigh symptoms/complaints (L14)	16 (5.4)	276.7 ± 42.8	4 (0.7)

GIST, gastrointestinal stromal tumor; GP, general practitioner; ICPC, International Classification of Primary Care.

**Table 3 cancers-17-01391-t003:** The most common newly prescribed drugs by the GP in the 12 months prior to GIST diagnosis (cases) and the index date (controls).

Drugs Prescribed 0–4 Months Prior to Diagnosis			
ATC code	ATC meaning	N(%)	N (%)
		GIST cases N = 294	Cancer-free controls N = 576
A06AD	Osmotically acting laxatives	44 (15.0)	5 (0.9)
A02BC	Proton pump inhibitors	41 (13.9)	15 (2.6)
A03FA	Propulsives	17 (5.8)	4 (0.7)
H02AB	Glucocorticoids	15 (5.1)	10 (1.7)
B01AC	Platelet aggregation inhibitors, excluding heparin	14 (4.8)	12 (2.1)
C07AB	Beta-blocking agents, selective	12 (4.1)	10 (1.7)
A06AB	Contact laxatives	11 (3.7)	1 (0.2)
B03AA	Iron bivalent oral preparations	10 (3.4)	0 (0)
J01CA	Penicillins with extended spectrum	10 (3.4)	9 (1.6)
N05BA	Benzodiazepine derivatives	9 (3.1)	4 (0.7)
Drugs prescribed ≤12–5 months prior to diagnosis			
ATC code	ATC meaning	N (%)	N (%)
A02BC	Proton pump inhibitors	83 (28.2)	97 (16.8)
C10AA	HMG CoA reductase inhibitors	65 (22.1)	117 (20.3)
C07AB	Beta-blocking agents, selective	45 (15.3)	77 (13.4)
B01AC	Platelet aggregation inhibitors, excluding heparin	44 (15.0)	68 (11.8)
C09AA	ACE inhibitors, plain	35 (11.9)	58 (10.1)
A06AD	Osmotically acting laxatives	32 (10.9)	41 (7.1)
C08CA	Dihydropyridine derivatives	29 (9.9)	46 (8.0)
C03AA	Thiazides, plain	27 (9.2)	36 (6.3)
C09CA	Angiotensin II receptor blockers (ARBs), plain	22 (7.5)	39 (6.7)
R03AC	Adrenergics for systemic use	20 (6.8)	5 (0.9)

## Data Availability

The data used in this study cannot be shared openly, as they are owned by the Nivel and the IKNL. For further inquiries regarding data access, please contact the corresponding author.

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
