# Peer review of "The Prediagnostic General Practitioners’ Pathway of Gastrointestinal Stromal Tumor Patients: A Real-World Data Study"

_cancers, 2025, doi:10.3390/cancers17091391_

Round 1
Reviewer 1 Report
Comments and Suggestions for Authors
Dear Authors,
First and foremost, I would like to commend the authors for their outstanding work on this paper. This study underscores the increased frequency of GP consultations prior to a GIST diagnosis, highlighting the potential role of primary care interactions in facilitating earlier detection.
My comments on this article are as follows:
1) Provide more details on the statistical methods used, such as the rationale for selecting the Wilcoxon rank-sum test and any adjustments made for multiple comparisons.​
2) Expand on the study's limitations, including potential biases in data collection, coding inconsistencies, and the retrospective nature of the analysis.​
3) Suggest specific strategies for integrating free-text GP notes and patient-reported outcomes in future research to enhance early detection methods.
Author Response
Dear reviewers,
We are pleased to be given the opportunity to revise our manuscript entitled “The Prediagnostic General Practitioners Pathway of Gastrointestinal Stromal Tumor Patients: a Real-World Data Study." with manuscript cancers-3531217. We would like to thank the reviewers for the review of our manuscript and the suggestions for revision. We are submitting two versions of the revised manuscript, one in which the revisions are highlighted with track changes, and one “clean” copy (the new version of the manuscript). We have also included a point-by-point response to the comments of the reviewer. We have incorporated all the recommended changes into the revised manuscript. The entire manuscript has also been reviewed by an English speaker, and I have implemented their suggestions.
We hope that the revised manuscript adequately addresses the issues raised.
Reviewer 1:
Dear Authors,
First and foremost, I would like to commend the authors for their outstanding work on this paper. This study underscores the increased frequency of GP consultations prior to a GIST diagnosis, highlighting the potential role of primary care interactions in facilitating earlier detection.
My comments on this article are as follows:
1) Provide more details on the statistical methods used, such as the rationale for selecting the Wilcoxon rank-sum test and any adjustments made for multiple comparisons.​
Answer to comment 1) First, thank you for your kind words and taking the time to review our article. We appreciate the opportunity to revise the manuscript.
The Wilcoxen rank-sum test is appropriate when comparing two independent groups (cases and controls) within a non-normally distributed dataset. In this study, we did not adjust for confounders, as this was not the primary aim. The revised method section now contains the following sentence: ‘This non-parametric test was chosen to examine differences between the two independent groups, and no adjustments were made for potential confounders, as this was not the primary aim.’ (page 4, line 165-166).
2) Expand on the study's limitations, including potential biases in data collection, coding inconsistencies, and the retrospective nature of the analysis.​
Answer to comment 2) Thank you for your comment. In the study’s limitations, we did not explicitly address the retrospective nature of the study and potential biases. We had briefly already mentioned coding inconsistencies on page 14, line 387 – we have now expanded on all these aspects. The updated limitations section includes (page 14, line 372-382): ‘The retrospective design of this study may have introduced some bias. However, we believe we selected a representative sample of the broader GIST population. Recall bias, which is more common in questionnaire-based studies, was likely minimal since the data was recorded either during or shortly after patient consultation. However, because primary care data is not collected for research purposes, it may be incomplete - a common limitation of RWD[21]. Furthermore, we were unable to review uncoded diagnoses and symptoms in the free text (i.e. clinical information expressed in free text, not restricted to predefined ICPC codes), which may have led to some underrepresentation of in primary care. Additionally, the sole use of primary care data including consultation frequency, diagnosis and prescribed drugs is not enough to map out the pre-diagnostic trajectory.’ (page 14, line 380).’
3) Suggest specific strategies for integrating free-text GP notes and patient-reported outcomes in future research to enhance early detection methods.
Answer to comment 3) We agree that suggesting specific strategies for integrating free-text GP notes adds value to the manuscript. We have now incorporated this into the manuscript as follows (page 14, line 404-408): ‘Natural language processing could help extract meaningful information from unstructured data and machine learning techniques could identify potential patterns and relationships within it. Furthermore, future studies should information on laboratory and diagnostic test results, as well as referral dates to secondary care, to provide a clearer picture of the pre-diagnostic trajectory [21,48]’.
Furthermore, we believe we have already specified how patient-reported outcomes could be utilized in future research (page 14, line 394). Hopefully this is clear enough.
We would like to thank you once more for your relevant comments and time.
Reviewer 2 Report
Comments and Suggestions for Authors
This is a well-designed study regarding presentation of patients with GIST compared to matched controls with regard to primary care visits preceding diagnosis. The underlying question is interesting and the aim of the paper- to help primary care providers better detect GIST, a rare tumor type- is admirable. I worry that specifying “GIST” and comparing controls to general population may not be as valuable as comparing “GIST” to patients diagnosed with other types of tumors like gastric adenocarcinoma or small bowel neuroendocrine tumor – or comparing all patients with GI tumors vs controls without GI tumors diagnosed.
Introduction
-Very clear, concise review of GIST and explains why this work is important, no comments on improvement
Methods
- Very clear. I wonder why the team decided to keep the patients who only had 1 matched control instead of 2. Were there any analysis performed after removing that small subgroup to ensure it wasn’t impacting the results in an outsize fashion? It shouldn’t, but it is curious.
Results
- Why not study the number of GP visits or symptoms comparing the cohort who carried “malignant neoplasm of stomach” to those who didn’t within the GIST case set? Seems like that would be a way to decide who was already in the workup phase and who was still a diagnostic mystery?
Discussion
-The first paragraph highlights a substantial limitation to the study – knowing that the visits occurred but not why they occurred- and points to the fact that not much can be concluded from the main finding of the study. This reinforces the opinion that although this was well designed, the utility of the paper is not as high as others.
The finding that “malignant GI tract neoplasm” is found more frequently for visits in patients with GIST is hardly remarkable – that seems to suggest that the GPs are working up and diagnosing the problem. Not sure this is a high priority finding and is highlighted in the conclusion.
Tables-
The tables display in a strange way – table 2 and 3 have identical columns for ICPC meanings for each control and GIST cases. This is unnecessarily redundant – one could remove the second column and both case types could refer to the first listed column alone.
Table 2 - I am not sure why the days between visit and diagnosis is documented just for the GIST patients and not the controls.
Author Response
Dear reviewers,
We are pleased to be given the opportunity to revise our manuscript entitled “The Prediagnostic General Practitioners Pathway of Gastrointestinal Stromal Tumor Patients: a Real-World Data Study." with manuscript cancers-3531217. We would like to thank the reviewers for the review of our manuscript and the suggestions for revision. We are submitting two versions of the revised manuscript, one in which the revisions are highlighted with track changes, and one “clean” copy (the new version of the manuscript). We have also included a point-by-point response to the comments of the reviewer. We have incorporated all the recommended changes into the revised manuscript. The entire manuscript has also been reviewed by an English speaker, and I have implemented their suggestions.
We hope that the revised manuscript adequately addresses the issues raised.
Reviewer 2:
This is a well-designed study regarding presentation of patients with GIST compared to matched controls with regard to primary care visits preceding diagnosis. The underlying question is interesting and the aim of the paper- to help primary care providers better detect GIST, a rare tumor type- is admirable. I worry that specifying “GIST” and comparing controls to general population may not be as valuable as comparing “GIST” to patients diagnosed with other types of tumors like gastric adenocarcinoma or small bowel neuroendocrine tumor – or comparing all patients with GI tumors vs controls without GI tumors diagnosed.
Introduction
-Very clear, concise review of GIST and explains why this work is important, no comments on improvement
Answer:
Thank you so much for your kind words, taking the time to read the article, and offering your valuable feedback.
To address your concern about the value of comparing GIST patients to the general population, rather than to patients with other tumor types (e.g. gastric adenocarcinoma or small bowel neuroendocrine tumors): the goal of our current analysis was to identify features that help distinguish GIST from individuals without cancer. In principle, someone with other tumor types (e.g. stomach cancer) would also be referred due to symptoms, so it is also relevant to compare GIST patients to the general population that are not referred to the hospital – as this reflects the diagnostic challenge in distinguishing GIST from individuals without cancer.
Comparing GIST to other tumor types would indeed provide different and valuable insights, but this is something we plan to explore in a follow-up paper.
Methods
- Very clear. I wonder why the team decided to keep the patients who only had 1 matched control instead of 2. Were there any analysis performed after removing that small subgroup to ensure it wasn’t impacting the results in an outsize fashion? It shouldn’t, but it is curious.
Answer to comment 1:
Thank you for your kind compliment and insightful question. Since our sample size was already relatively small (N=294), we decided to include all patients who had at least 1 control, including the 12 who had only one matched control instead of two. We examined whether there were any differences between patients with one control versus two and found them to be similar in terms of age and sex. The fact that these patients only have one control is likely because they come from smaller GP practices in the Netherlands, making it more difficult to find two controls for these cases.
Results
- Why not study the number of GP visits or symptoms comparing the cohort who carried “malignant neoplasm of stomach” to those who didn’t within the GIST case set? Seems like that would be a way to decide who was already in the workup phase and who was still a diagnostic mystery?
Answer to comment 2:
Thank you for your question. The cases with the code “malignant neoplasm of stomach” were not necessarily in the workup phase during diagnosis. As discussed in this article, the high number of recorded diagnosis of malignant neoplasms of the GI tract likely reflect instances where ICPC codes were updated as more clinical information became available (see page 12, line 309-317). Thus, we do not know whether these patients were already in the workup phase at that time or whether their records were updated to malignant neoplasm codes when more clinical information became available. From this reason, comparing these cases to those without this code would not be meaningful. We hope this explanation is clear, but please let us know if you need further clarification.
Discussion
-The first paragraph highlights a substantial limitation to the study – knowing that the visits occurred but not why they occurred- and points to the fact that not much can be concluded from the main finding of the study. This reinforces the opinion that although this was well designed, the utility of the paper is not as high as others.
Answer to comment 3:
Thank you for your comment. We agree that this is a limitation of the study; however, we believe it remains important to present the current findings. Our aim was to gain insights into the primary care pathway for this patient group, and we see this study as a valuable first step towards that understanding. We agree that future research is needed, as we mention in the discussion (page 14, line 394) and that some conclusions - such as the reasons for the observed increase in visits - cannot be drawn from this study alone. We also mention this in the discussion in the revised version of the manuscript (page 14, line 380): ‘Additionally, the usage of solely primary care date including consultation frequency, diagnosis and prescribed drugs is not enough to map out the pre-diagnostic trajectory.’. However, we believe we emphasize this point in both the discussion and conclusion. Despite this, we feel the utility of this paper remains high, as it provides important initial insights that are currently lacking in literature. Additionally, this study design is not limited to GIST patients in primary care and could be applied to other cancer types as well. And in future research, combining different types of data, such as patient-reported reasons for visiting the GP, could help to provide an even more complete picture.
The finding that “malignant GI tract neoplasm” is found more frequently for visits in patients with GIST is hardly remarkable – that seems to suggest that the GPs are working up and diagnosing the problem. Not sure this is a high priority finding and is highlighted in the conclusion.
Answer to comment 4:
Many thanks for your comment. Like we mention in our answer to comment 2, the cases with the code “malignant neoplasm of stomach” were not necessarily in the workup phase during diagnosis. In the conclusion we state which codes were most seen prior to diagnosis (Diagnoses related to malignant GI tract neoplasms, various types of abdominal pain, and iron deficiency anemia) however, we also discuss that it is of relevance to study the investigate initial symptoms that were coded during consultations as it might be these codes were updated when more clinical information became available (page 14, line 417).
Tables-
The tables display in a strange way – table 2 and 3 have identical columns for ICPC meanings for each control and GIST cases. This is unnecessarily redundant – one could remove the second column and both case types could refer to the first listed column alone.
Answer to comment 5:
Many thanks for this relevant suggestion. We totally agree with you. See the updated table 2 and table 3 on page 7 and page 17.
Table 2 - I am not sure why the days between visit and diagnosis is documented just for the GIST patients and not the controls.
Answer to comment 6:
Thank you for your question. The reason is that the controls do not have a diagnosis date; instead, they were assigned an index date (page 4, line 138), which matches the diagnosis date of their corresponding cases, to allow for the retrieval of their symptoms and diagnosis. Therefore, we believe it is not relevant to also show this information for the controls. We hope this is clear, if not, please let us know.
Reviewer 3 Report
Comments and Suggestions for Authors
The manuscript by Dr. Holthuis et al examines visits to general practitioners prior to diagnosis with GIST, finding that patients had more frequent GP visits in the year before a GIST diagnosis.
Major critiques
- The analysis is limited to 1 country and database. Generalizability to the US and other countries with very different primary and specialty care structures is unclear. This is especially true for countries where many people have more limited access to care and where universal health care is not a reality. However, nothing that can really be done to fix this critique other than including datasets outside of the Netherlands, though this is outside the scope of the study, not readily attainable, and not necessary for publication of this manuscript.
Minor critiques
- Introduction
- Line 76, add “whether” so it reads “depending on whether adjuvant imatinib is given.”
- Line 79-80, recommend replacing “ulcer-like symptoms” with “dyspepsia.”
- Line 82, remove “GISTs”
- Line 91, change “potentially” to “potential”
- Line 96, remove the comma
- Methods
- Lines 133-134 – using this nomenclature (Tx, T2, etc) may confuse readers if they are not paying careful attention because we use these terms for TNM staging. Please revise here and throughout the manuscript.
- Results
- Line 171 – what does this 9,8% refer to? Percentage of what group?
- Are there any data available on use of EGD and/or colonoscopy in the 1-2 years prior to diagnosis?
- Discussion
- Lines 265-266 – why couldn’t you compare across GIST locations? I thought you were able to distinguish between gastric and non-gastric? This would be ideal to see in some of the comparisons made in the Results section. For example, are patients with eventual gastric GIST diagnosis more likely to have more GP visits prior to diagnosis than non-gastric?
- Line 296 – remove “with”
- I would like the authors to speculate/pontificate more about potential areas of intervention that could be pursued to hasten GIST diagnosis in patients presenting to GPs for the reasons that GIST patients present in months prior to eventual diagnosis. For example, would earlier EGD and/or imaging be helpful? Could this be a population where eventually a liquid biopsy (e.g. ctDNA) could be helpful for diagnosis?
- Tables/Figures
- Error bars are missing in the bar graphs in Figures 2 and A1.
Comments on the Quality of English Language
See above detailed comments.
Author Response
Dear reviewers,
We are pleased to be given the opportunity to revise our manuscript entitled “The Prediagnostic General Practitioners Pathway of Gastrointestinal Stromal Tumor Patients: a Real-World Data Study." with manuscript cancers-3531217. We would like to thank the reviewers for the review of our manuscript and the suggestions for revision. We are submitting two versions of the revised manuscript, one in which the revisions are highlighted with track changes, and one “clean” copy (the new version of the manuscript). We have also included a point-by-point response to the comments of the reviewer. We have incorporated all the recommended changes into the revised manuscript. The entire manuscript has also been reviewed by an English speaker, and I have implemented their suggestions.
We hope that the revised manuscript adequately addresses the issues raised.
Reviewer 3:
Major critiques
- The analysis is limited to 1 country and database. Generalizability to the US and other countries with very different primary and specialty care structures is unclear. This is especially true for countries where many people have more limited access to care and where universal health care is not a reality. However, nothing that can really be done to fix this critique other than including datasets outside of the Netherlands, though this is outside the scope of the study, not readily attainable, and not necessary for publication of this manuscript.
Answer to comment 1:
We would like to thank you for taking the time to read the manuscript and provide your valuable feedback.
We agree that the results are limited to countries with comparable healthcare systems. However, several countries, including the United Kingdom, Scandinavian nations and Australia, share similar healthcare structures. For future research, it would indeed also be interesting to conduct such a study in a different healthcare system, such as in the US.
As you pointed out, this limitation is beyond the scope of the manuscript, but we do mention this as a limitation in the discussion. Following your comment, we have expanded this limitation to: ‘Additionally, our findings may not be generalizable to healthcare systems where access to care is more limited and where universal healthcare is not available.’ (page 14, line 385).
We would also like to mention that we plan to conduct a similar study in Australia and the United Kingdom, with a different study population.
Minor critiques
- Introduction
- Line 76, add “whether” so it reads “depending on whether adjuvant imatinib is given.”
- Line 79-80, recommend replacing “ulcer-like symptoms” with “dyspepsia.”
- Line 82, remove “GISTs”
- Line 91, change “potentially” to “potential”
- Line 96, remove the comma
Answer to minor critiques:
Thank you very much for your suggestions. We have incorporated these suggestion into the manuscript at the corresponding lines except for suggestion 3, line 82, remove ‘’GISTs’’. GISTs has been retained at the start of the sentence for clarity and to avoid confusion with the subject of the preceding sentence.
- Methods
- Lines 133-134 – using this nomenclature (Tx, T2, etc) may confuse readers if they are not paying careful attention because we use these terms for TNM staging. Please revise here and throughout the manuscript.
Answer to comment 3:
We agree that this could be confusing for readers. To address this, we have revised the manuscript accordingly, like you suggested, and decided to replace Tx with Time 1, Time 2 and so on. Many thanks for this relevant comment.
- Results
- Line 171 – what does this 9,8% refer to? Percentage of what group?
Answer to comment 4:
This number is based on the individuals in the linked Nivel-PCD and NCR study sample, from which we only included those diagnosed with GIST between 2010 and 2020 and diagnosis of GIST and who had a corresponding control. However, after carefully reviewing the text, we have decided omit the percentage, as it did not add significant value and was potentially confusing to readers.
- Are there any data available on use of EGD and/or colonoscopy in the 1-2 years prior to diagnosis?
Answer to comment 5:
Many thanks for your question. Unfortunately, we do not have data on the use of EGD or colonoscopy, as our study was limited to primary care data, which does not include these procedures (in the Netherlands). In the discussion (page 13, line 369), we acknowledge that such information, along with referral data, would be valuable for future research to get a completer idea on the diagnostic trajectory. However, for this study, our focus was specifically on events within primary care, making these data outside the scope of our analysis.
- Discussion
- Lines 265-266 – why couldn’t you compare across GIST locations? I thought you were able to distinguish between gastric and non-gastric? This would be ideal to see in some of the comparisons made in the Results section. For example, are patients with eventual gastric GIST diagnosis more likely to have more GP visits prior to diagnosis than non-gastric?
Answer to comment 6:
I appreciate your question. In this study, we did not compare GIST locations due to the small sample sizes (Gastric GIST n= 196 & Gastrointestinal GIST, other (excl. stomach) n=98), which we believe are insufficient for drawing meaningful conclusions. However, we do agree with you that it would be interesting in future studies to compare across GIST locations and see if differences exist in presenting symptoms as well as GP consultation frequency.
- Line 296 – remove “with”
Answer to comment 7:
Thank you for your suggestion. We believe that the sentence on line 296 is grammatically correct: we also confirmed this with a native English speaker. We hope it is acceptable to leave it as it is.
- I would like the authors to speculate/pontificate more about potential areas of intervention that could be pursued to hasten GIST diagnosis in patients presenting to GPs for the reasons that GIST patients present in months prior to eventual diagnosis. For example, would earlier EGD and/or imaging be helpful? Could this be a population where eventually a liquid biopsy (e.g. ctDNA) could be helpful for diagnosis?
Answer to comment 8:
Thank you for your comment and suggestion. In this study, we focused exclusively on primary care data, and such tests are not performed in primary care in the Netherlands. Furthermore, we did not have the necessary data to investigate this, and it falls outside the scope of this article. However, we agree that this would be an interesting avenue for future research. Based on your comment and others, we have expanded the discussion section to include: ‘Additionally, the sole use of primary care data including consultation frequency, diagnosis and prescribed drugs is not enough to map out the pre-diagnostic trajectory.’ (page 14, line 380) and in the future section that ‘Furthermore, future studies should information on laboratory and diagnostic test results, as well as referral dates to secondary care, to provide a clearer picture of the pre-diagnostic trajectory [21,48]’. We hope this sufficiently addresses your concern, and we appreciate your valuable input.
- Tables/Figures
- Error bars are missing in the bar graphs in Figures 2 and A1.
Answer to comment 9:
Thank you for this comment. We have now included the error bars in the bar graphs, Figure 2 and Figure A1 (page 8 and 17).
Reviewer 4 Report
Comments and Suggestions for Authors
This is a very interesting study on patients history before the diagnosis of a GIST based on the frequency of GP consultations. However, the increased consultations of the GP are just an observation and cannot help to detect deficiencies in the diagnostic process of GIST. For quality control, the time of referral to relevant diagnostic procedures like CT and endoscopy are mandatory. This data are missing, as the authors state. This is a problem of the study design. However, without this data no concrete conclusions are possible.
Why did the authors not include the time point of referral to diagnostic procedures? Reasons should be addressed in the Methods section and this shortcoming needs much more addressed in the discussion.
Of course, the best would be to include this data in the manuscript
Author Response
Dear reviewers,
We are pleased to be given the opportunity to revise our manuscript entitled “The Prediagnostic General Practitioners Pathway of Gastrointestinal Stromal Tumor Patients: a Real-World Data Study ." with manuscript cancers-3531217. We would like to thank the reviewers for the review of our manuscript and the suggestions for revision. We are submitting two versions of the revised manuscript, one in which the revisions are highlighted with track changes, and one “clean” copy (the new version of the manuscript). We have also included a point-by-point response to the comments of the reviewer. We have incorporated all the recommended changes into the revised manuscript. The entire manuscript has also been reviewed by an English speaker, and I have implemented their suggestions.
We hope that the revised manuscript adequately addresses the issues raised.
Reviewer 4:
This is a very interesting study on patients history before the diagnosis of a GIST based on the frequency of GP consultations. However, the increased consultations of the GP are just an observation and cannot help to detect deficiencies in the diagnostic process of GIST. For quality control, the time of referral to relevant diagnostic procedures like CT and endoscopy are mandatory. This data are missing, as the authors state. This is a problem of the study design. However, without this data no concrete conclusions are possible.
Why did the authors not include the time point of referral to diagnostic procedures? Reasons should be addressed in the Methods section and this shortcoming needs much more addressed in the discussion.
Of course, the best would be to include this data in the manuscript
Answer:
First, we would like to thank you for taking the time to read this article and your kind compliments.
We agree that the observed increase in GP visits is merely an observation and this study does not allow us to determine whether or how the diagnostic trajectory can be improved at the primary care level. We discuss this in the discussion and conclusion (page 12, line 301 and page 15, line 419).
As you pointed out, we did not have data on the time to referral for relevant diagnostic procedures. Ideally, future studies should integrate multiple sources of real-world data to provide a more comprehensive understanding of the (pre-)diagnostic trajectory of patients. In response to your feedback, we have further acknowledged this in the manuscript. The methods section now states: ‘Data on referral to secondary care, was well as relevant diagnostic procedures such as CT and endoscopy, were not available.’ (page x, line x). and in the discussion section ‘We also lacked data on lifestyle or environmental risk factors for GISTs, as well as referrals to secondary care or diagnostic procedures, which could provide further insights into diagnostic pathways and potential delays.’ (page 13, line 369) and ‘Furthermore, future studies should information on laboratory and diagnostic test results, as well as referral dates to secondary care, to provide a clearer picture of the pre-diagnostic trajectory.’ (page 14, line 406).
We appreciate your valuable input and hope this revision addresses your concerns.
Round 2
Reviewer 4 Report
Comments and Suggestions for Authors
Thank you for your revised manuscript. I am aware, that you cannot analyse reasons of possible delays in diagnosis due to a lack of data. However, the manuscript points that out much clearer and I understand it as a first step for assessing the quality of the diagnostic process.
No further critics